

# No one-size-fits-all solution to clean GBIF

Alexander Zizka[1,2], Fernanda Antunes Carvalho[3], Alice Calvente[4], Mabel Rocio Baez-Lizarazo[5], Andressa Cabral[6], Jéssica Fernanda Ramos Coelho[4], Matheus Colli-Silva[6], Mariana Ramos Fantinati[4], Moabe F. Fernandes[7], Thais Ferreira-Araújo[4], Fernanda Gondim Lambert Moreira[4], Nathália Michelly da Cunha Santos[4], Tiago Andrade Borges Santos[7], Renata Clicia dos Santos-Costa[4], Filipe C. Serrano[8], Ana Paula Alves da Silva[4], Arthur de Souza Soares[4], Paolla Gabryelle Cavalcante de Souza[4], Eduardo Calisto Tomaz[4], Valéria Fonseca Vale[4], Tiago Luiz Vieira[7] and Alexandre Antonelli[9,10,11]

[1] sDiv, German Centre for Integrative Biodiversity Research Halle-Jena-Leipzig (iDiv), Leipzig, Germany
[2] Naturalis Biodiversity Center, Leiden, The Netherlands
[3] Departamento de Genética, Ecologia e Evolução, Universidade Federal de Minas Gerais, Belo Horizonte, Brazil
[4] Departamento de Botânica e Zoologia, Universidade Federal do Rio Grande do Norte, Natal, Brazil
[5] Departamento de Botânica, Universidade Federal do Rio Grande do Sul, Porto Alegre, Brazil
[6] Departamento de Botânica, Universidade de São Paulo, São Paulo, Brazil
[7] Departamento de Ciências Biológicas, Universidade Estadual de Feira de Santana, Feira de Santana, Brazil
[8] Departamento de Ecologia, Universidade de São Paulo, São Paulo, Brazil
[9] Gothenburg Global Biodiversity Centre, University of Gothenburg, Gothenburg, Sweden
[10] Department for Biological and Environmental Sciences, University of Gothenburg, Gothenburg, Sweden
[11] Royal Botanic Gardens Kew, Richmond, United Kingdom

Corresponding author
Alexander Zizka,
alexander.zizka@idiv.de

## ABSTRACT

Species occurrence records provide the basis for many biodiversity studies. They derive from georeferenced specimens deposited in natural history collections and visual observations, such as those obtained through various mobile applications. Given the rapid increase in availability of such data, the control of quality and accuracy constitutes a particular concern. Automatic filtering is a scalable and reproducible means to identify potentially problematic records and tailor datasets from public databases such as the Global Biodiversity Information Facility (GBIF; http://www.gbif.org), for biodiversity analyses. However, it is unclear how much data may be lost by filtering, whether the same filters should be applied across all taxonomic groups, and what the effect of filtering is on common downstream analyses. Here, we evaluate the effect of 13 recently proposed filters on the inference of species richness patterns and automated conservation assessments for 18 Neotropical taxa, including terrestrial and marine animals, fungi, and plants downloaded from GBIF. We find that a total of 44.3% of the records are potentially problematic, with large variation across taxonomic groups (25–90%). A small fraction of records was identified as erroneous in the strict sense (4.2%), and a much larger proportion as unfit for most downstream analyses (41.7%). Filters of duplicated information, collection year, and basis of record, as well as coordinates in urban areas, or for terrestrial taxa in the sea or marine taxa on land, have the greatest effect. Automated filtering can help in identifying problematic records, but requires customization of which tests and thresholds should be applied to the taxonomic group and geographic area under focus. Our results stress the importance

of thorough recording and exploration of the meta-data associated with species records for biodiversity research.

## INTRODUCTION

Publicly available species distribution data have become a crucial resource in biodiversity research, including studies in ecology, biogeography, systematics and conservation biology. In particular, the availability of digitized collections from museums and herbaria and citizen science observations has increased drastically over the last few years. As of today, the largest public aggregator for geo-referenced species occurrences data, the Global Biodiversity Information Facility (http://www.gbif.org), provides access to more than 1.5 billion geo-referenced occurrence records for species from across the globe and the tree of life.

A central challenge to the use of these publicly available species occurrence data in research is problematic geographic coordinates, which are either erroneous or unfit for downstream analyses (for instance because they are overly imprecise, *Anderson et al., 2016*). Problems mostly arise because data aggregators such as GBIF integrate records collected with different methodologies in different places at different times—often without centralized curation and only rudimentary meta-data. For instance, problematic coordinates caused by data-entry errors or automated geo-referencing from vague locality descriptions are common (*Maldonado et al., 2015*; *Yesson et al., 2007*) and cause recurrent problems such as records of terrestrial species in the sea, records with coordinates assigned to the centroids of political entities, or records of species in cultivation or captivity (*Zizka et al., 2019*).

Manual data cleaning based on expert knowledge can detect these issues, but it is only feasible on small taxonomic or geographic scales, and it is time-consuming and difficult to reproduce. As an alternative, automated filtering methods to identify potentially problematic records have been proposed as a scalable option, as they are able to deal with datasets containing up to millions of records and many different taxa. Those methods are usually based on geographic gazetteers (e.g., *Chamberlain, 2016*; *Zizka et al., 2019*; *Jin & Yang, 2020*) or on additional data, such as environmental variables (*Robertson, Visser & Hui, 2016*). Additionally, filtering procedures based on record meta-data, such as collection year, record type, and coordinate precisions, have been proposed to improve the suitability of publicly available occurrence records for biodiversity research (*Zizka et al., 2019*).

Problematic records are especially critical in conservation, where stakes are high. Recently proposed methods for automated conservation assessments could support the formal assessment procedures for the global Red List of the International Union for the Conservation of Nature (IUCN) (*Dauby et al., 2017*; *Bachman et al., 2011*; *Pelletier et al., 2018*). These methods approximate species' range size, namely the Extent of Occurrence

(EOO, which is the area of a convex hull polygon comprising all records of a species), the Area of Occupancy (AOO, which is the sum of the area actually occupied by a species, calculated based on a small-scale regular grid), and the number of locations for a preliminary conservation assessment following IUCN Criterion B ("Geographic range"). These methods have been used to propose preliminary global (*Stévart et al., 2019*; *Zizka et al., 2020*) and regional (*Schmidt et al., 2017*; *Cosiaux et al., 2018*) Red List assessments. However, all metrics, and especially EOO, are sensitive to individual records with problematic coordinates. Automated conservation assessments may therefore be biased, particularly if the number of records is low, as it is the case for many tropical species.

While automated filters hold great promise for biodiversity research, their use across taxonomic groups and datasets remains poorly explored. Here, we test the effect of automated filtering of species geographic occurrence records on the number of records available in different groups of animals, fungi, and plants. Furthermore, we test the impact of automated filtering procedures for the accuracy of preliminary automated conservation assessments compared to full IUCN assessments. Specifically, we evaluate a pipeline of 13 automated filters to flag possibly problematic records by using record meta-data and geographic gazetteers in two categories: (1) erroneous (coordinates, that are likely wrong, irrespective of the downstream analyses, for instance due to data entry errors) and (2) unfit for purpose (coordinates that are not wrong per se, but likely unfit for the planned downstream analyses, for instance because they are overly imprecise). We address three questions:

1. Which filters lead to the biggest loss of data when applied?
2. Does the importance of individual filters differ among taxonomic groups?
3. Does automated filtering improve the accuracy of automated conservation assessments?

## MATERIAL AND METHODS

### Choice of study taxa

This study is the outcome of a workshop held at the Federal University of Rio Grande do Norte in Natal, Brazil in October 2018 which gathered students and researchers working with different taxonomic groups of animals, fungi, and plants across the Neotropics (Fig. 1). Each participant analysed geographic occurrence data from their taxonomic group of interest and commented on the results for their group. Hence, we include groups based on the expertise of the participants rather than following an arbitrary choice of taxa and taxonomic ranks. We acknowledge a varying degree of documented expertise and number of years working on each group. We obtained public occurrence records for 18 taxa, including one plant family, nine plant genera, one genus of fungi, three families and one genus of arthropods, one family of snakes, one family of skates, and one genus of bony fish (Table 1).

### Species occurrence data

We downloaded occurrence information for all study groups from http://www.gbif.org using the `rgbif` v1.4.0 package (*Chamberlain, 2017*) in R (*GBIF.org, 2019a*; *GBIF.org,*

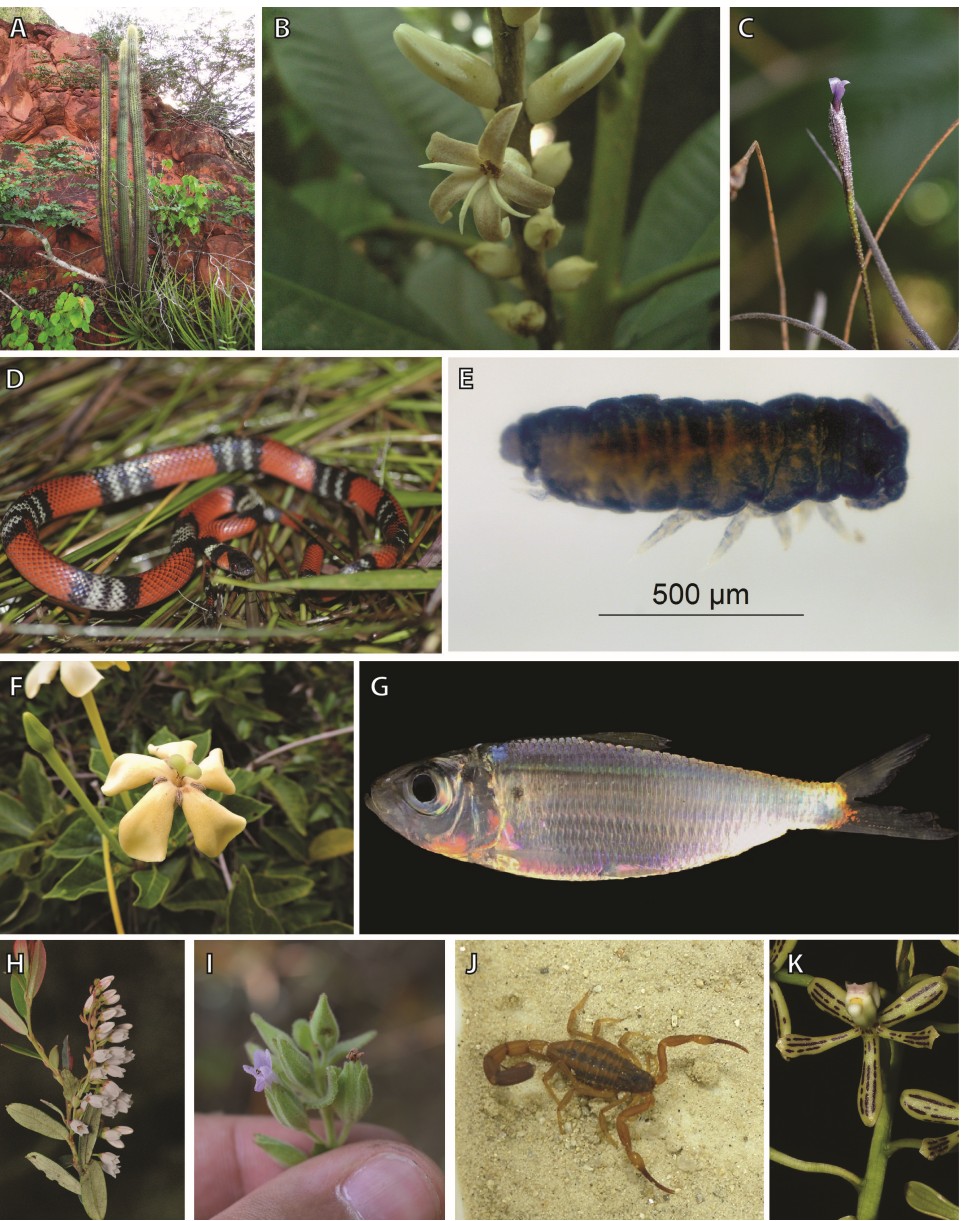

**Figure 1** **Examples of taxa included in this study.** (A) *Pilosocereus pusillibaccatus* (*Pilosocereus*), (B) *Conchocarpus macrocarpus* (*Conchocarpus*); (C) *Tillandsia recurva* (*Tillandsia*); (D) *Oxyrhopus guibei* (Dipsadidae); (E) *Aethiopella ricardoi* (Neanuridae); (F) *Tocoyena formosa* (*Tocoyena*); (G) *Harengula jaguana* (*Harengula*); (H) *Gaylussacia decipiens* (*Gaylussacia*); (I) *Oocephalus foliosus* (*Oocephalus*); (J) *Tityus carvalhoi* (*Tityus*); (K) *Prosthechea vespa* (*Prosthechea*), Image credits: (A) Pamela Lavor, (B) Juliana El-Ottra, (C) Eduardo Calisto Tomaz, (D) Filipe C. Serrano, (E) Raiane Vital da Paz (available under a Creative Commons Attribution 3.0 Unported License), (F) Fernanda G.L. Moreira, (G) Thais Ferreira-Araujo, (H) Luiz Menini Neto, (I) Arthur de Souza Soares, (J) Renata C. Santos-Costa, (K) Tiago Vieira.

*2019b*; *GBIF.org, 2019c*; *GBIF.org, 2019d*; *GBIF.org, 2019e*; *GBIF.org, 2019f*; *GBIF.org, 2019g*; *GBIF.org, 2019h*; *GBIF.org, 2019i*; *GBIF.org, 2019j*; *GBIF.org, 2019k*; *GBIF.org, 2019l*; *GBIF.org, 2019m*; *GBIF.org, 2019n*; *GBIF.org, 2019o*; *GBIF.org, 2019p*; *GBIF.org, 2020a*;

**Table 1  The study groups and their taxonomy.** This study includes three marine and 15 terrestrial taxa, seven of them animals, one group of fungi and ten plants, belonging to 16 different orders. The outlines illustrate the broad taxonomic group (i.e., an evolutionary relative if relative if no icon of the direct study group was available). Icons from http://www.phylopic.org if available under a Public Domain license otherwise created by the authors (*Heath, 2020*; *Hillewaert, 2006*; *Hough, 2008*; *Menchetti, 2020*; *McNair, 2020a*; *McNair, 2020b*; *Müller, 1885*; *Nimphel, 2020*; *Petar, 2020*; *Pohl, 1827*; *Reinke, 2020*; *PhyloPic, 2020a*; *PhyloPic, 2020b*; *Welter-Schultes, 2017*; *Xgirouxb, 2020*; *Veronidae, 2012*).

| | Taxon | Taxon rank | Realm | Common name | 'Phylum' | Family |
|---|---|---|---|---|---|---|
| | Diogenidae | Family | Marine | Hermit crabs | Arthropoda | Diogenidae |
| | Entomobryidae | Family | Terrestrial | Springtails | Arthropoda | Entomobryidae |
| | Neanuridae | Family | Terrestrial | Springtails | Arthropoda | Neanuridae |
| | *Tityus* | Genus | Terrestrial | Scorpions | Arthropoda | Buthidae |
| | Arhynchobatidae | Family | Marine | Skates | Chordata | Arhynchobatidae |
| | Dipsadidae | Family | Terrestrial | Snakes | Chordata | Dipsadidae |
| | *Harengula* | Genus | Marine | Herrings | Chordata | Clupeidae |
| | *Thozetella* | Genus | Terrestrial | Sac fungi | Ascomycota | Chaetosphaeriaceae |
| | *Conchocarpus* | Genus | Terrestrial | NA | Angiosperms | Rutaceae |
| | *Gaylussacia* | Genus | Terrestrial | Huckleberries | Angiosperms | Ericaceae |
| | *Harpalyce* | Genus | Terrestrial | NA | Angiosperms | Fabaceae |
| | Iridaceae | Family | Terrestrial | NA | Angiosperms | Iridaceae |
| | *Lepismium* | Genus | Terrestrial | Cacti | Angiosperms | Cactaceae |
| | *Oocephalus* | Genus | Terrestrial | NA | Angiosperms | Lamiaceae |
| | *Pilosocereus* | Genus | Terrestrial | Cacti | Angiosperms | Cactaceae |
| | *Prosthechea* | Genus | Terrestrial | Orchids | Angiosperms | Orchidaceae |
| | *Tillandsia* | Genus | Terrestrial | Bromeliads | Angiosperms | Bromeliaceae |
| | *Tocoyena* | Genus | Terrestrial | NA | Angiosperms | Rubiaceae |

*GBIF.org, 2020b*). We downloaded GBIF-interpreted data including only records with geographic coordinates and limited the study area to a rectangle between 90°S–33°N and 35°W–120°W reflecting the Neotropics (*Morrone, 2014*), our main area of expertise. The natural distributions of all included taxa are confined to the Neotropics except for Arhynchobatidae, Diogenidae, Dipsadidae, Entomobryidae, *Gaylussacia*, Iridaceae, Neanuridae, and *Tillandsia*, for which we only obtained the Neotropical occurrences. We

consider GBIF data generally of high quality and use them as a case study because GBIF is the largest, most widely used and taxonomically most comprehensive data source for species occurrence records; however many more exist (e.g., https://bien.nceas.ucsb.edu/bien/, http://www.fishbase.de or *Guedes et al., 2018*). GBIF provides information on the internal consistency of records, among others including information on decimal rounding of coordinates, geographic projection, date validity and geospatial issues. Since we specifically aimed to test the effect of user-level filtering we included records flagged with issues by GBIF (this was also the default option). Geospatial issues flagged by GBIF only concerned 0.4% of the records used in this study and including them had the added benefit to make our results directly comparable to other databases, which may use different internal consistency checks or none at all.

## Automated cleaning

We followed the cleaning pipeline outlined by *Zizka et al. (2019)* and first filtered the data as downloaded from GBIF ("raw", hereafter) using meta-data for those records for which they were available (although meta-data were often missing, *Peterson et al., 2018*), removing: (1) records with a coordinate precision below 100 km (as this represents the grain size of many macro-ecological analyses); (2) fossil records and records of unknown source; (3) records collected before 1945 (before the end of the Second World War, since coordinates of old records are often imprecise); and (4) records with an individual count of less than one and more than 99. Furthermore, we rounded the geographic coordinates to four decimal places and retained only one record per species per location (i.e., test for duplicated records). In a second step, we used the `clean_coordinates` function of the `CoordinateCleaner v2.0-11` package (*Zizka et al., 2019*) with default options to flag errors that are common to biological data sets ("filtered", hereafter). These include: coordinates in the sea for terrestrial taxa and on land for marine taxa, coordinates containing only zeros, coordinates assigned to country and province centroids, coordinates within urban areas, and coordinates assigned to biodiversity institutions. See Table 2 for a summary of all filters we used and their classification into "erroneous" and "unfit".

## Downstream analyses

We first generated species richness maps using 100x100 km grid cells for the raw and filtered datasets respectively, using the package `speciesgeocodeR v2.0-10` (*Töpel et al., 2017*). We then performed an automated conservation assessment for all study groups based on both datasets using the `ConR v1.2.4` package (*Dauby et al., 2017*). ConR estimates the EOO, AOO, and the number of locations, and then suggests a preliminary conservation status based on Criterion B of the global IUCN Red List. While these assessments are preliminary (see *IUCN Standards and Petitions Subcommittee, 2017*), they can be a proxy used by the IUCN to speed up full assessments. We then benchmarked the preliminary conservation assessments against the global IUCN Red List assessments for the same taxa (where available), which we obtained from http://www.iucn.org via the `rredlist v.0.5.0` package (*Chamberlain, 2018*).

**Table 2  The automated filters used in this study.**

| Test | Type | Basis | Rationale |
|---|---|---|---|
| Biodiversity institutions | Error | Gazetteer-based | Records may have coordinates at the location of biodiversity institutions, e.g., because they were erroneously entered with the physical location of the specimen or because they represent individuals from captivity or horticulture, which were not clearly labeled as such |
| Equal lat/lon | Error | Gazetteer-based | Coordinates with equal latitude and longitude are usually indicative of data entry errors |
| Sea | Error | Gazetteer-based | Coordinates from terrestrial organisms in the sea are usually indicative of data entry errors, e.g., swapped latitude and longitude |
| Zeros | Error | Gazetteer-based | Coordinates with plain zeros are often indicative of data entry errors |
| Capitals | Unfit | Gazetteer-based | Records may be assigned to the coordinates of country capitals based on a vague locality description |
| Duplicates | Unfit | Gazetteer-based | Duplicated records may add unnecessary computational burden, in particular for large scale biodiversity analyses and distribution modelling for many species |
| Political centroids | Unfit | Gazetteer-based | Records may be assigned to the coordinates of the centroids of political entities based on a vague locality description |
| Urban areas | Unfit | Gazetteer-based | Records from urban areas are not necessarily errors, but often represent imprecise records automatically geo-referenced from vague locality descriptions or old records from different land-use types |
| Basis of record | Unfit | Meta-data | Records might be unsuitable or unreliable for certain analyses dependent on their source, e.g., 'fossil' or 'unknown' |
| Collection year | Unfit | Meta-data | Coordinates from old records are more likely to be imprecise or erroneous coordinates since they are derived from geo-referencing based on the locality description. This is more problematic for older records, since names or borders of places may change |
| Coordinate precision | Unfit | Meta-data | Records may be unsuitable for a study if their precision is lower than the study analysis scale |
| Identification level | Unfit | Meta-data | Records may be unsuitable if they are not identified to species level. |
| Individual count | Unfit | Meta-data | Records may be unsuitable if the number of recorded individuals is 0 or if the count is too high. This may be related to data-entry or data-basing problems (e.g., defaulting to 0 for numerical values), indicate records from DNA barcoding and in some cases indicate records of absence. |

## Evaluation of results

Each author provided an informed comment on the performance of the raw and cleaned datasets, concerning the number of removed records and the accuracy of the overall species richness maps. We then compared the agreement between automated conservation assessments based on raw and filtered occurrences with the global IUCN Red List for those taxa where IUCN assessments were available (http://www.iucn.org).

We carried out all analyses in the R computing environment (*R Core Team, 2019*), using standard libraries for data handling and visualization (*Wickham, 2018*; *Garnier, 2018*; *Ooms, 2014*; *Ooms, 2019*; *Hijmans, 2019*). All scripts are available from a Zenodo repository (doi:10.5281/zenodo.3695102).

## RESULTS

We retrieved a total of 218,899 species occurrence records, with a median of 2,844 records per study group and 10 records per species (Table 3, Appendix S1). We obtained most records for Dipsadidae (64,249) and fewest for *Thozetella* (51). The species with most records was *Harengula jaguana* (19,878).

Our automated tests filtered a total of 97,004 records (Fig. 2, erroneous: 9,254, unfit: 91,298), with a median of 45% per group (erroneous: 0.3%, unfit: 37.4%). Overall, the most important test was for duplicated records (on average 35.5% per taxonomic group). The filtering steps based on record meta-data that filtered the largest number of records were the basis of records (5.9%) and the collection year (3.4%). The most important automated tests were for urban area (8.6%) and the occurrence from records of terrestrial taxa in the sea and marine taxa on land (4.3%, see Table 3 and Appendix S1 in the electronic supplement for further details and the absolute numbers). Only a few records were filtered by the coordinate precision, zero coordinates and biodiversity institution tests (Fig. 3).

Entomobryidae, Diogenidae, and Neanuridae had the highest fraction of filtered records (Table 3). In general, the different filters we tested were of similar importance for different study groups. There were few outstanding exceptions, including the particularly high proportions of records filtered by the "basis of record test" for *Tityus* (7.0%), Dipsadidae (5.6%), *Prosthechea* (5.0%) and *Tillandsia* (4.9%), by the collection year for Dipsadidae (11.3%), by the taxonomic identification level for *Tityus* (1.6%), by the capital coordinates for *Oocephalus* (6.1%) and *Gaylussacia* (3.2%), by the seas/land test for Diogenidae and *Thozetella*, and by the urban areas test for *Oocephalus* (13.3%) and Iridaceae (12.3%). Furthermore, Entomobryidae differed considerably from all other study taxa with exceptionally high numbers of records filtered by the "basis of record", "level of identification" and "urban areas" tests.

Geographically, the records filtered by the "basis of record" and "individual count" tests were concentrated in Central America and southern North America, and a relatively high number of records were filtered due to their proximity to the centroids of political entities were located on Caribbean islands (Fig. 3). See Appendix S2 for species richness maps using the raw and cleaned data for all study groups.

We found IUCN assessments for 579 species that were also included in our distribution data from 11 of our study groups (Table 4, Appendix S3). The fraction of species evaluated varied among the study group, with a maximum of 100% for *Harengula* and *Lepismium* and a minimum of 2.3% for Iridaceae (note that the number of total species varied considerably among groups). The median percentage of species per study group with an IUCN assessment was 15%. A total of 102 species were listed as *Threatened* by the IUCN global Red List (CR = 19, EN = 40, VU = 43) and 477 as *Not Threatened*.

Zizka et al. (2020), *PeerJ*, DOI 10.7717/peerj.9916

**Table 3  The impact of automated filtering on occurrence records for 18 Neotropical taxa downloaded from http://www.gbif.org.**  From column six onwards the numbers show the percentage of records flagged by the respective test. Only tests that flagged at least 0.1% of the records in any group are shown. Individual records can be flagged by multiple tests, therefore the sum of percentages from all tests can supersede the total percentage.

| Taxon | Summary | | | | Errors | | | Unfit | | | | | | | | |
| --- | --- | --- | --- | --- | --- | --- | --- | --- | --- | --- | --- | --- | --- | --- | --- | --- |
| | Total records | Fraction flagged [%] | Fraction error [%] | Fraction unfit [%] | Biodiversity Institutions [%] | Sea/land area [%] | Zeros [%] | Capitals [%] | Duplicates [%] | Political centroids [%] | Urban areas [%] | Basis of record [%] | Collection year [%] | Coordinate precision [%] | Id-level [%] | Individual count [%] |
| Diogenidae | 13,840 | 68.7 | 44.3 | 38.2 | 0.0 | 44.3 | 0.0 | 0.7 | 33.8 | 0.2 | 1.3 | 1.7 | 2.5 | 0.0 | 0.0 | 0.0 |
| Entomobryidae | 2,767 | 90.3 | 0.1 | 90.3 | 0.1 | 0.0 | 0.0 | 0.1 | 85.5 | 0.0 | 70.1 | 72.9 | 2.0 | 0.0 | 72.1 | 0.0 |
| Neanuridae | 689 | 66.9 | 0.0 | 66.9 | 0.0 | 0.0 | 0.0 | 0.0 | 62.4 | 0.0 | 2.0 | 2.9 | 1.3 | 0.0 | 0.0 | 0.0 |
| *Tityus* | 1,018 | 55.2 | 0.5 | 54.9 | 0.5 | 0.0 | 0.0 | 1.2 | 43.5 | 0.1 | 6.9 | 7.0 | 0.4 | 1.8 | 1.6 | 0.0 |
| Arhynchobatidae | 14,633 | 38.5 | 3.8 | 37.4 | 0.0 | 3.8 | 0.0 | 0.0 | 35.4 | 0.0 | 1.9 | 1.7 | 1.3 | 0.0 | 0.9 | 0.0 |
| Dipsadidae | 64,249 | 57.7 | 0.3 | 57.6 | 0.3 | 0.0 | 0.0 | 1.8 | 46.3 | 0.4 | 8.5 | 5.6 | 11.3 | 0.8 | 0.0 | 0.1 |
| *Harengula* | 36,697 | 31.0 | 5.5 | 27.8 | 0.0 | 5.5 | 0.0 | 0.2 | 27.0 | 0.1 | 0.2 | 1.0 | 0.4 | 0.0 | 0.3 | 0.0 |
| *Thozetella* | 51 | 35.3 | 23.5 | 29.4 | 0.0 | 23.5 | 0.0 | 0.0 | 27.5 | 0.0 | 2.0 | 0.0 | 0.0 | 0.0 | 0.0 | 0.0 |
| *Conchocarpus* | 1,551 | 43.2 | 0.5 | 42.9 | 0.1 | 0.4 | 0.0 | 0.0 | 39.6 | 0.9 | 2.3 | 0.5 | 1.9 | 0.1 | 0.0 | 0.0 |
| *Gaylussacia* | 3,998 | 47.2 | 0.1 | 47.1 | 0.1 | 0.1 | 0.0 | 3.2 | 41.8 | 1.1 | 5.2 | 0.7 | 4.4 | 0.6 | 0.0 | 0.0 |
| *Harpalyce* | 870 | 33.1 | 0.0 | 33.1 | 0.0 | 0.0 | 0.0 | 1.0 | 26.0 | 1.3 | 3.8 | 0.5 | 5.5 | 0.7 | 0.0 | 0.9 |
| Iridaceae | 23,127 | 33.6 | 0.5 | 33.5 | 0.4 | 0.1 | 0.0 | 1.0 | 17.1 | 0.4 | 12.3 | 0.9 | 4.7 | 0.1 | 0.0 | 1.3 |
| *Lepismium* | 825 | 29.7 | 0.0 | 29.7 | 0.0 | 0.0 | 0.0 | 0.1 | 21.9 | 0.1 | 7.8 | 0.0 | 2.1 | 0.0 | 0.0 | 0.0 |
| *Oocephalus* | 883 | 49.3 | 0.0 | 49.3 | 0.0 | 0.0 | 0.0 | 6.1 | 41.9 | 0.8 | 13.3 | 0.0 | 0.7 | 0.3 | 0.0 | 0.1 |
| *Pilosocereus* | 1,940 | 25.9 | 0.2 | 25.9 | 0.2 | 0.0 | 0.0 | 0.5 | 16.8 | 0.5 | 2.1 | 1.8 | 7.0 | 0.0 | 0.0 | 0.9 |
| *Prosthechea* | 6,617 | 31.5 | 0.1 | 31.5 | 0.0 | 0.0 | 0.1 | 0.4 | 19.6 | 1.7 | 0.9 | 5.0 | 8.3 | 0.1 | 0.0 | 0.2 |
| *Tillandsia* | 42,222 | 35.3 | 0.4 | 35.2 | 0.3 | 0.0 | 0.0 | 0.7 | 19.8 | 0.7 | 9.2 | 4.9 | 5.1 | 0.1 | 0.0 | 1.0 |
| *Tocoyena* | 2,922 | 37.6 | 0.3 | 37.4 | 0.0 | 0.2 | 0.0 | 0.8 | 32.3 | 0.8 | 5.0 | 0.1 | 1.9 | 0.2 | 0.0 | 0.5 |
| Total | 218,899 | 44.3 | 4.2 | 41.7 | 0.2 | 4.0 | 0.0 | 1.0 | 32.3 | 0.4 | 7.1 | 4.2 | 5.6 | 0.3 | 1.0 | 0.4 |

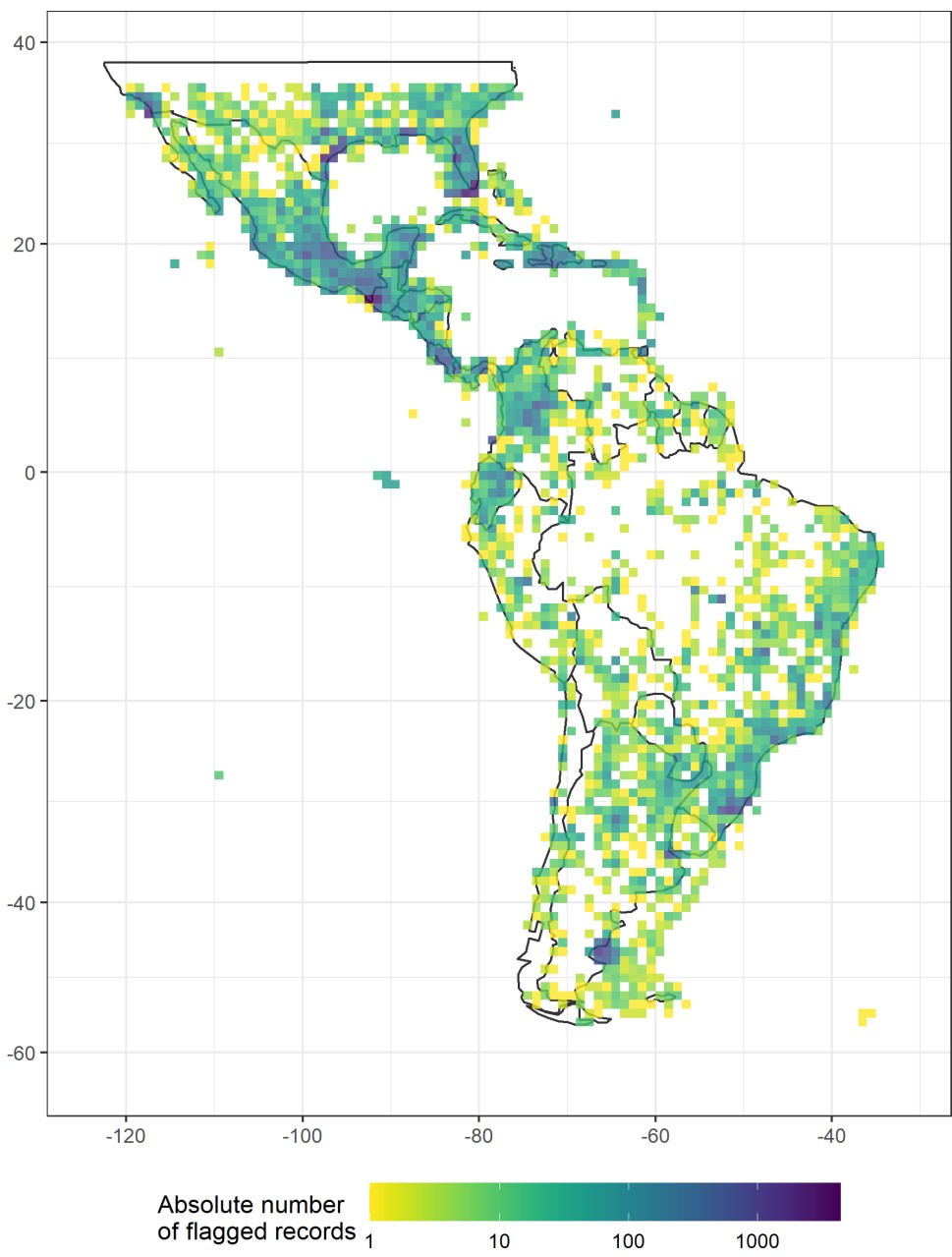

**Figure 2** The absolute number of records flagged as erroneous or unfit by automated geographic filters in a dataset of 18 Neotropical taxa including animals, fungi, and plants, plotted in a 100 × 100 km grid across the Neotropics (Behrmann projection).

We obtained automated conservation assessments for 2,181 species in the filtered dataset. Based on the filtered data, the automated conservation assessment evaluated 1,382 species as possibly threatened (63.4%, CR = 495, EN = 577, VU = 310, see Appendix S3 for assessments of all species). The automated assessment based on the filtered dataset agreed with the IUCN assessment for identifying species as possibly threatened (CR, EN,

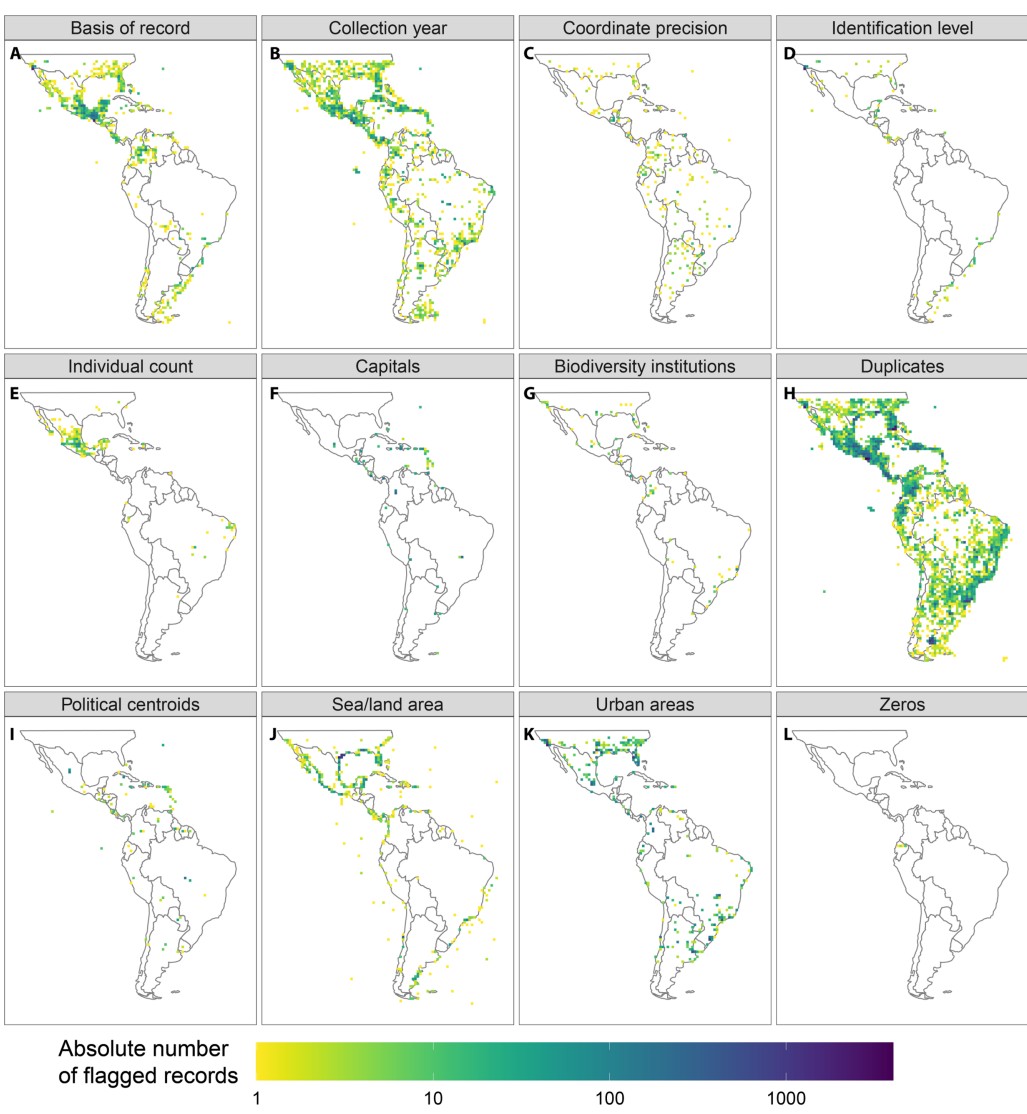

**Figure 3 Geographic location of the occurrence records flagged by the automated tests applied in this study.** Only filters that flagged at least 0.1% of records in any taxon are shown. (A) Basis of records, (B) Collection year, (C) Coordinate precision, (D) Identification level, (E) Individual count, (F) Capitals, (G) Biodiviersity Institutions, (H) Duplicates, (I) Political centroids, (J) Sea/land area, (K) Urban areas, (L) Zeros.

VU) for 358 species (64%; Table 4). Filtering reduced the EOO by 18.4% and the AOO by 9.9% on median per group. For the raw dataset the agreement with IUCN was higher at 381 species (65.7%).

## DISCUSSION

Automated flagging based on meta-data and automatic tests filtered on average 45% of the records per taxonomic group; 25.9%–90.3% as "unfit" and 0%–44.3% as "erroneous". The filters for basis of record, duplicates, collection year, and urban areas flagged the

Zizka et al. (2020), *PeerJ*, DOI 10.7717/peerj.9916

**Table 4  Conservation assessment for 11 Neotropical taxa of plants and animals based on three datasets.** IUCN: global Red List assessment obtained from http://www.iucn.org; GBIF Raw: preliminary conservation assessment based on IUCN Criterion B using ConR and the raw dataset from GBIF; GBIF filtered: preliminary conservation assessment based on IUCN Criterion B using ConR and the filtered dataset. Only taxa with at least one species evaluated by IUCN shown.

| Taxon | IUCN | | | GBIF Raw | | | GBIF Filtered | | | | |
|---|---|---|---|---|---|---|---|---|---|---|---|
| | *n* taxa | Evaluated [%] | Threatened [%] | *n* taxa | Threatened [%] | Match with IUCN [%] | *n* taxa | Threatened [%] | Match with IUCN [%] | EOO change compared to raw [%] | AOO change compared to raw [%] |
| Arhynchobatidae | 37 | 51.3 | 17.9 | 39 | 35.9 | 45.0 | 39 | 41.0 | 40.0 | −32.7 | −18.5 |
| Dipsadidae | 520 | 68.0 | 8.8 | 638 | 58.3 | 63.0 | 598 | 59.9 | 61.2 | −2.3 | −15.6 |
| *Harengula* | 4 | 100.0 | 0.0 | 4 | 0.0 | 100.0 | 4 | 0.0 | 100.0 | −38.0 | −36.9 |
| *Conchocarpus* | 4 | 8.7 | 0.0 | 46 | 63.0 | 100.0 | 45 | 62.2 | 100.0 | −15.3 | −7.1 |
| *Gaylussacia* | 2 | 3.3 | 0.0 | 61 | 59.0 | 50.0 | 58 | 60.3 | 50.0 | −22.5 | −8.6 |
| *Harpalyce* | 3 | 15.0 | 5.0 | 20 | 65.0 | 66.7 | 17 | 58.8 | 50.0 | −18.4 | −16.5 |
| Iridaceae | 13 | 2.3 | 0.2 | 531 | 64.4 | 50.0 | 466 | 62.9 | 62.5 | −18.2 | −12.3 |
| *Lepismium* | 6 | 100.0 | 0.0 | 6 | 16.7 | 83.3 | 6 | 16.7 | 83.3 | −33.9 | −7.9 |
| *Pilosocereus* | 41 | 80.9 | 19.1 | 47 | 55.3 | 73.7 | 46 | 56.5 | 71.1 | −8.5 | −5.8 |
| *Tillandsia* | 54 | 11.6 | 6.0 | 464 | 61.4 | 85.2 | 453 | 62.7 | 83.3 | −13.7 | −9.9 |
| *Tocoyena* | 3 | 13.6 | 4.5 | 22 | 31.8 | 66.7 | 21 | 38.1 | 66.7 | −23.0 | −9.5 |

highest fraction of records (**Question 1**). The importance of different tests was similar across taxonomic groups, with particular exceptions for the tests on basis of record, collection year, capital coordinates, and urban areas (**Question 2**). The results for species richness were similar between the raw and filtered data with some improvements using the filters. We found little impact of filtering on the accuracy of the automated conservation assessments **Question 3**).

## The relevance of individual filters

The aim of automated filtering is to identify possibly problematic records that are unsuitable for particular downstream analyses. While those records filtered as "erroneous" will likely cause problems for most biodiversity research, those filtered as "unfit" might have varying impact, depending on the type and spatial resolution of the downstream analyses. Unwanted effects include an unnecessary computational burden, which can be a bottleneck for large-scale analyses (i.e., duplicates, *Antonelli et al., 2018*), and increased uncertainty (due to low precision), or completely compromising results. For instance, records assigned to country centroids might be acceptable for inter-continental comparisons, but are likely to be erroneous for species distribution modelling on a local scale. The importance of each test and the linked thresholds must be judged based on the specific downstream analyses. As our results show, it may be advisable to adapt automated tests to the geographic study area or the taxonomic study group. For instance, the high number of records flagged for centroids on the Lesser Antilles (Fig. 3) might be overly strict (https://data-blog.gbif.org/post/country-centroids/), although we chose a conservative distance for the Political centroid test (1 km).

Several factors may explain the high proportion of records flagged as duplicates. First, the deposition of duplicates from the same specimen at different institutions is common practice, especially for plants, where a specimen duplication is entirely feasible. Second, independent collections at similar localities may occur, in particular for local endemics. Third, low coordinate precision, for instance based on automated geo-referencing from locality descriptions, may lump records from nearby localities. Fourth, different data contributors might add the same record to GBIF, if their sources overlap, as can for instance be the case for the Barcode of Life and Plazi databases.

## Similarities and differences among taxa

The number of records flagged by individual tests was similar across study groups, suggesting that similar problems might be relevant for collections of plants and animals. Therefore, the same filters can be used across taxonomic groups. Some notable exceptions stress the need to adapt each filter to the taxonomic study group to balance data quality and data availability. The high fraction of records filtered by the "basis of record" filter for *Tityus*, Dipsadidae, *Prosthechea* and *Tillandsia*, were mostly caused by a high number of records in these groups based on unknown collection methods, which might be caused by the contribution of specific datasets lacking this information for these groups. The high fraction of records flagged by the "collection year" filter for Dispadidae was caused by a high collection effort in the late 1880s and early 1900s, as can be expected for a charismatic

group of reptiles, but also by 500 records dated to the year 1700. The latter records likely represent a data entry error: they are all contributed to GBIF from the same institution, and the institution's code for unavailable collection dates is 1700-01-01–2014-01-01, which has likely erroneously been converted to 1700. The high number of species flagged at capital coordinates and within urban areas for the plant groups Iridaceae and *Oocephalus* might be related to horticulture, since at least some species in those groups are commonly cultivated as ornamentals. This was supported by the detailed examination of the data for Iridaceae, which showed that after filtering 1605 records from 69 exotic species remained in the dataset, stressing the importance to address these species in certain taxonomic groups.

The general agreement between the species richness maps based on raw and filtered data was encouraging, in terms of the use of this data for large-scale biogeographic research (Fig. 4, Appendix S2). The filter based on political centroids had an important impact on species richness patterns, which is congruent with the results from a previous study in the coffee family (*Maldonado et al., 2015*). Records assigned to country or province centroids are often old records, which are geo-referenced at a later point based on vague locality descriptions. These records are at the same time more likely to represent dubious species names, since they might be old synonyms or type specimens of species that have only been collected and described once, which are erroneously increasing species numbers.

Overall, we consider the effect of the automated filters as positive since they identified the above-mentioned issues and increased the data precision and reduced computational burden (Table 3, Appendix S2). However, in some cases filters failed to remove major issues, often due to incomplete meta-data. For instance, for Diogenidae we found at least two records of an species known only from Eocene fossils (*Paguristes mexicanus*) which slipped the "basis of record" test because they were marked as "preserved specimen" rather than "fossil specimen". Furthermore, for Entomobryidae we found that for 1,996 records the meta-data on taxonomic rank was "UNRANKED" despite all of them being identified to species level, leading to a high fraction of records removed by the "Identification level" filter. Additionally automated filters might be overly strict or unsuitable for certain taxa. For instance, in Entomobryidae, 2,004 samples were marked as material samples and therefore removed by our global filter retaining only specimen and observation data, which in this case was overly strict.

The filters we included in this study address a set of important but relatively easy to identify problems. In fact, the internal quality control of GBIF does flag some of the problems we tested for (i.e., zero coordinates, equal lat/lon) while others might be implemented in the near future (country centroids, https://data-blog.gbif.org/post/country-centroids/). While this internal quality control is very helpful, we see a huge potential to overcome issues with data quality in a user-feedback system that allows users to provide expert assessments, i.e., a meta-annotation of records being challenged (and why). Such a system would not need to change the original data and could include multiple levels to account for differing opinions.

As next steps for automated filtering, tests for intrinsic consistency and support by external data (if available) can help to detect additional problematic records. For instance, testing if records' coordinates fall within the state or province of collection noted for a

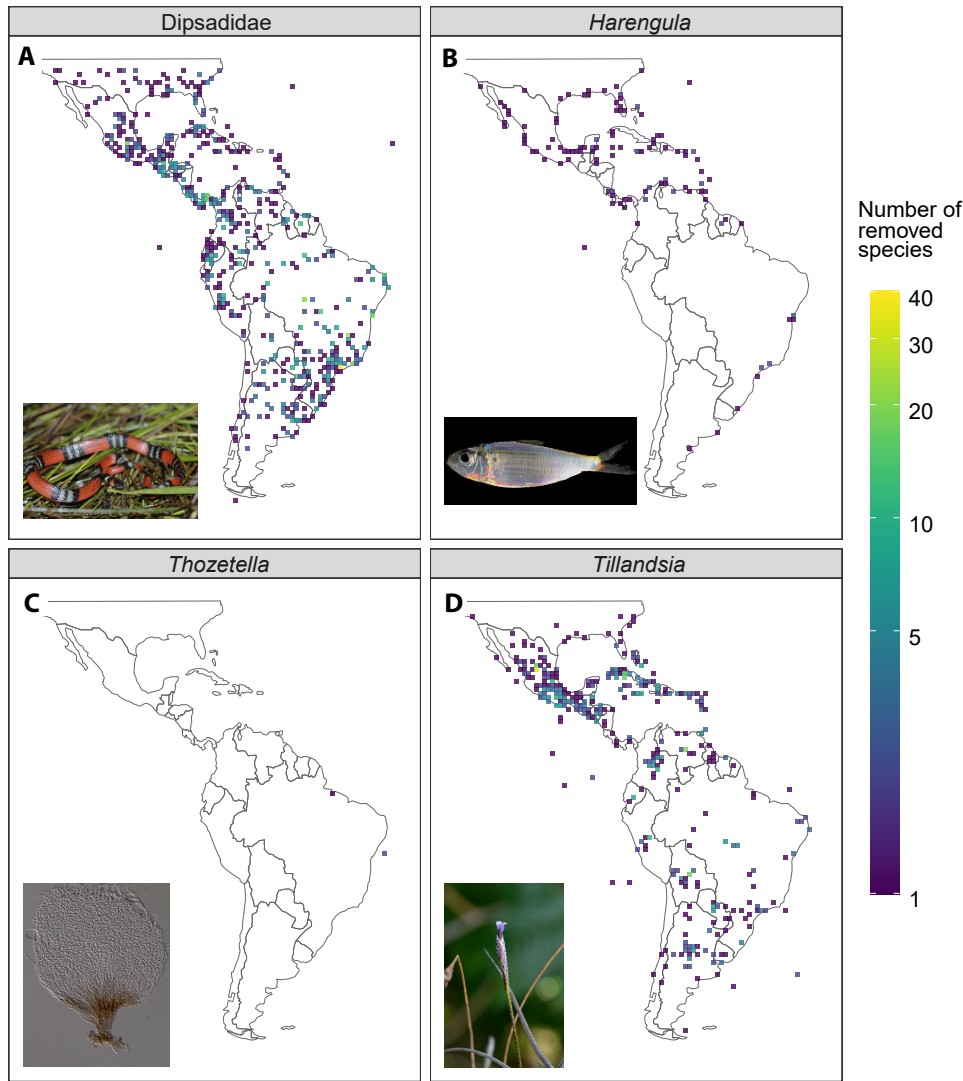

**Figure 4** **Illustrative examples of the difference in species richness between the raw and filtered dataset (raw - filtered) from four of the study taxa.** (A) Dipsadidae (Total number of species in the dataset, $n = 637$), (B) *Harengula* ($n = 4$), (C) *Thozetella* ($n = 9$), (D) *Tillandsia* ($n = 464$). Photo credits for (C) by Tiago Andrade Borges Santos, otherwise as in Fig. 1.

record (intrinsic) or testing if they agree with external species distribution information, for example from http://www.iucn.org (vertebrates; extrinsic) or https://wcsp.science.kew.org/ (selected seed plant families; extrinsic) can further corroborate the accuracy of a record's geographic referencing. If such tests are included, it is essential to account for the sampling year, in particular for older records, since the names of political entities may change and the ranges of species may shift. Furthermore, while in this study we focused on meta-data and geographic filtering, taxonomic cleaning—the resolution of synonymies and identification of accepted names—is another important part of data curation, but

depends on taxon-specific taxonomic backbones and synonymy lists which are not readily available for many groups and often are contradictory within individual taxa.

### The impact of filtering on the accuracy of automated conservation assessments

The accuracy of the automated conservation assessment was in the same range as found by previous studies (*Nic Lughadha et al., 2019*; *Zizka et al., 2020*). The similar accuracy of the raw and filtered dataset for the automated conservation assessment was surprising, in particular given the EOO and AOO reduction observed in the filtered dataset (Table 4) and the impact of errors on spatial analyses observed in previous studies (*Gueta & Carmel, 2016*). The robustness of the automated assessment was likely due to the fact that the EOO for most species was large, even after the considerable reduction caused by filtering. This might be caused by the structure of our comparison, which only included species that were evaluated by the IUCN Red List (and not considered as *Data Deficient*) and at the same time had occurrences recorded in GBIF. Those inclusion criteria are likely to have biased the datasets towards species with large ranges, since generally more data are available for them. The robustness of automated conservation assessments to data quality is encouraging, although these methods are only an approximation (and not replacements) of full IUCN Red List assessments, especially for species with few collection records (*Rivers et al., 2011*).

## CONCLUSIONS

Our results suggest that between one quarter to half of the occurrence records obtained from GBIF might be unsuitable for downstream biodiversity analyses. While the majority of these records might not be erroneous *per se*, they are overly imprecise and thereby increase uncertainty of downstream results or add computational burden on big data analyses.

While our results suggest that large-scale species richness patterns and automated conservation assessments are largely resilient to the effects of problematic occurrence records, they also stress the importance of (meta-)data exploration prior to most biodiversity analyses. Automated filtering can help to identify problematic records, but also highlight the necessity to customize tests and thresholds to the specific taxonomic groups and geographic area of interest. The putative problems we encountered point to the importance to train researchers and students to curate species occurrence datasets and to visibly associate user-feedback with individual records on aggregator platforms such as GBIF so that it can contribute to the overall accuracy and precision of public biodiversity databases.

## ACKNOWLEDGEMENTS

We thank GBIF and all data collectors and contributors for their excellent work. We thank Town Peterson, Roderic Page and two anonymous reviewers for the helpful comments on an earlier version of this manuscript. This study enrolled participants of the workshop "Biodiversity data: from field to yield" led by Alice Calvente, Fernanda Carvalho, Alexander

Zizka, and Alexandre Antonelli through the Programa de Pós Graduação em Sistemática e Evolução of the Universidade Federal do Rio Grande do Norte (UFRN) and promoted by the 6th Conference on Comparative Biology of Monocotyledons - Monocots VI.

### Funding

This research was funded by the Pró-reitoria de Pesquisa and the Pró-reitoria de Pós-graduação of UFRN (edital 02/2016 –internacionalização), iDiv via the German Research Foundation (DFG FZT 118), specifically through sDiv, the Synthesis Centre of iDiv, the Coordenação de Aperfeiçoamento de Pessoal de Nível Superior - Brasil (CAPES), Fundação de Amparo à Pesquisa do estado de São Paulo (FAPESP, process 2015/20215-7), the Swedish Research Council, the Knut and Alice Wallenberg Foundation, the Swedish Foundation for Strategic Research and the Royal Botanic Gardens, Kew. The funders had no role in study design, data collection and analysis, decision to publish, or preparation of the manuscript.

### Grant Disclosures

The following grant information was disclosed by the authors:
Pró-reitoria de Pesquisa and the Pró-reitoria de Pós-graduação of UFRN.
German Research Foundation: DFG FZT 118.
Coordenação de Aperfeiçoamento de Pessoal de Nível Superior - Brasil (CAPES) Fundação de Amparo à Pesquisa do estado de São Paulo: Process 2015/20215-7.
Swedish Research Council, the Knut and Alice Wallenberg Foundation.
The Swedish Foundation for Strategic Research and the Royal Botanic Gardens, Kew.

### Competing Interests

The authors declare there are no competing interests.

### Author Contributions

- Alexander Zizka conceived and designed the experiments, performed the experiments, analyzed the data, prepared figures and/or tables, authored or reviewed drafts of the paper, and approved the final draft.
- Fernanda Antunes Carvalho, Alice Calvente and Alexandre Antonelli conceived and designed the experiments, authored or reviewed drafts of the paper, and approved the final draft.
- Mabel Rocio Baez-Lizarazo, Andressa Cabral, Jéssica Fernanda Ramos Coelho, Matheus Colli-Silva, Mariana Ramos Fantinati, Moabe F. Fernandes, Thais Ferreira-Araújo, Fernanda Gondim Lambert Moreira, Nathália Michelly da Cunha Santos, Tiago Andrade Borges Santos, Renata Clicia dos Santos-Costa, Filipe C. Serrano, Ana Paula Alves da Silva, Arthur de Souza Soares, Paolla Gabryelle Cavalcante de Souza, Eduardo Calisto Tomaz, Valéria Fonseca Vale and Tiago Luiz Vieira performed the experiments, analyzed the data, authored or reviewed drafts of the paper, and approved the final draft.

## Data Availability

Code is available at GitHub (https://github.com/idiv-biodiversity/effects_of_automated_cleaning) and Zenodo: Zizka, Alexander, Antunes Carvalho, Fernanda, Calvente, Alice, Baez-Lizarazo, Mabel Rocio, Cabral, Andressa, Coelho, Jéssica Fernanda Ramos, Colli-Silva, Matheus, …Alexandre Antonelli. (2020). No one-size-fits-all solution to clean GBIF. http://doi.org/10.5281/zenodo.3695102.

## Supplemental Information

Supplemental information for this article can be found online at http://dx.doi.org/10.7717/peerj.9916#supplemental-information.

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
