# Peer review of "No one-size-fits-all solution to clean GBIF"

_PeerJ, doi:10.7717/peerj.9916_

## Round 0.1 · original submission · Major Revisions

The three referees are positive about the value of the paper but have some significant points they feel need correcting. In particular, it is essential to distinguish between what are true errors or simply facts of how data may be published through different sources; duplicate records are not errors when they come from different datasets for example. Authors often mistakenly use the phrase data 'cleaning' which suggests the data has mistakes. Rather, what they are really doing is selecting the data that is 'fit' for their 'purpose. We look forward to your careful revision of the paper.

Reviewer 1 ·

Basic reporting

L36-37: Suggest add a citation in "Here, we evaluate the effect of 13 recently proposed filters on the inference of species"

Experimental design

L102: Please clarify why you did not choose to add filters when accessing data from GBIF. In particular "hasGeospatialIssue", a subset of basis of record, and a year range would have removed records before downloading impacting the % results. The equal lat/lng and zero tests are carried out in GBIF.org processing.

L106: Please clarify what you mean by "raw". The GBIF downloads include verbatim data as provided by the institutions and interpreted data but it is unclear which were used.

L294-328: All citations are incorrect in lines 294-328 with a typing error (doi.org repeated twice).

Validity of the findings

L192: Recommend add a fourth reason that the data are shared through multiple databases - e.g. BOLD or Plazi may publish the same data as the institution

Additional comments

This is nicely written with clear accessible language.

I would like to see mention of data cleaning performed in GBIF.org before downloading (zero's, lat/lng test) which were seemingly overlooked, allowing you to emphasize the importance of the other aspects, such as country centroids which GBIF doesn't test for today.

You may refer to the country centroid blog (https://data-blog.gbif.org/post/country-centroids) and the gridded data blog (https://data-blog.gbif.org/post/gridded-datasets-update/) noting that GBIF intends to include automated tests to detect those issues.

·

Basic reporting

The manuscript is quite clean and readable, and I very much enjoyed reading it. There were a couple of English gaffs in it, but really very few, so I have no complaints at all in terms of basic reporting.

"We downloaded only records with geographic coordinates and limited the study area to latitudes smaller than 33◦ N and longitudes smaller than 35◦ W and larger than 120◦ 101 W reflecting the Neotropics" ... this is a little convoluted.

"as this represent the grain size of many macro-ecological analyses" ... should be representS"

The maps have a too-bold and too-black coastal outline, and need country boundaries.

The lit cited is not pretty

Experimental design

I appreciated the explicit reporting of study findings. The methods were clear, based on tools that are in the public domain, and as such the study appears to be robust and repeatable. Indeed, I also very much liked the "community effort" feel of this manuscript, in that it was a combination of taxonomic experts with analytical specialists. Very nice contribution.

Validity of the findings

I am convinced that the findings reported are robust. My only set of concerns with the manuscript is as regards the depth to which the data cleaning approaches were implemented. That is to say, the authors implemented a very basic set of tools, but I did not see a next level of tools, in which one would seek intrinsic and extrinsic consistency in data records. For instance, for each group, are there synonyms represented among the names, and do the names match consistently to one authority list. Or for geographic referencing, do the coordinates reported fall within the administrative boundaries of the higher-level areas reported (e.g., states, countries). These ideas of intrinsic and extrinsic consistency lend considerable additional support to a data record, and can be implemented rather easily in R. This does not necessarily need to be added, but could be mentioned as next possible steps.

My apologies for the self-promotion, but a paper in which I was involved might be relevant to this analysis also ... Peterson, A. T., A. Asase, D. A. L. Canhos, S. de Sousa, and J. Wieczorek. 2018. Data leakage and loss in biodiversity informatics. Biodiversity data journal 6:e26826.

Additional comments

In general, a nice contribution! All the best, Town Peterson

·

Basic reporting

This manuscript reports on a workshop where the participants evaluated issues with the quality of data obtained from GBIF, relative to their own taxonomic expertise. The report reads well. I have some minor suggestions.

Great to see the GBIF downloads cited with DOIs. However the URLs are incorrect ("doi.org" is duplicated). It would be helpful if each one was labelled by the corresponding taxon name. I found it frustrating having to try several times before I go specific taxa I was interested in.

Table 1 could be made more visually interesting by inserting images of the taxa at the start (e.g., from http://phylopic.org). Fig. 1 shows nice example of the taxa, but this is disconnected from the data presentation.

Experimental design

See below under comments on "Validity of the findings"

Validity of the findings

My overall sense is that the authors somewhat overstate their conclusions. The headline (line 38) of “29-90% of records are potentially erroneous” gets rolled back in the body of the manuscript. As they note later in the article (line 192) “Duplicated records do not represent errors *per se*”. Indeed, they are standard practise in botanical collections. The “basis of record” test doesn’t handle DNA sequence-based records (e.g., from DNA barcoding) which are typically recorded as “MATERIAL SAMPLE” (e.g., numerous records for the Entomobryidae). Issues with sea versus land (e.g., Diogenidae) seem in part to be due to coarse georeferencing taking a locality as being terrestrial rather than costal. For example, GBIF occurrence https://www.gbif.org/occurrence/699114012 comes from the MCZ (https://mczbase.mcz.harvard.edu/guid/MCZ:IZ:CRU-12772) and is a record from the Thayer Expedition to Brazil (see http://biostor.org/reference/4104) from “Victoria”. I suspect this is actually “Vitória, Espírito Santo”, a city on the coast of Brazil visited by that expedition.

Hence I suggest that what we have is a series of idiosyncratic data handling issues across a range of datasets. It’s not clear to me that these are errors. Indeed, the fact that these issues made little difference to the how well the automated assessments matched those by the IUCN could be taken as evidence that those methods are robust to these “errors”. Perhaps the conclusions are really that each taxon and datasets may be subject to various (sometimes taxon or data specific) issues, but these don’t have a major impact on their utility.

Additional comments

As I note in the section "Validity of the findings" I'm not convinced that many of the issues are, indeed, "errors". There are some taxon-specific practices (e.g., duplicates in botany), some egregious data handling issues (e.g., the date range for the snake specimens), and flagging DNA barcoding records on a "basis of record" term that seems perfectly valid. There's not much evidence that these "errors" actually impact the utility of the data in this case.

Ultimately I think the more interesting topic is where do these data issues come from, and how do we fix them so that future researchers benefit from data where these problems have been removed.

---

## Round 0.2 · Minor Revisions

Thank you for addressing the referees comments. They and one new referee are happy with the MS and make a few minor suggestions they feel would improve the paper.

·

Basic reporting

Acceptable.

Experimental design

Acceptable.

Validity of the findings

Acceptable.

Additional comments

I liked this manuscript from the outset, and the authors responded appropriately with revisions to meet my requests in my first review. My only remaining concern is in regard to the comment about further data-cleaning efforts that could be implemented and that should be mentioned. The authors responded to the *extrinsic* point (e.g., comparisons of data from GBIF with extrinsic range map summaries. They did not, however, respond to my suggestions of using *intrinsic* consistency as a further filter of data quality. For instance, one can compare the administrative unit in which the geographic coordinates associated with the data record fall with the administrative unit given as the country, state, or county in the data record. This consistency can be a useful corroboration of the geographic referencing... and such consistency assessments can be extremely useful in detecting data problems. So it is not all extrinsic, as the authors interpreted, but there are also intrinsic dimensions in which their assessments of data quality could be expanded. Other than that--i.e., simply mentioning these possibilities among future thoughts and possible avenues--I am quite happy with this manuscript. Town Peterson

·

Basic reporting

no comment

Experimental design

no comment

Validity of the findings

no comment

Additional comments

I'm happy with the changes made, this is an interesting manuscript.

I have one minor quibble. In Table 1 on p. 7 the caption says "Pictures from www.phylopic.org, in the public domain." This is not the case, or at least, not for the one image I checked. The hermit crab image http://phylopic.org/image/6b0a5129-4184-46ca-a7b8-e38838e12194/ is CC-NC-SA which requires attribution - "You must give credit to Ekaterina Kopeykina (vectorized by T. Michael Keesey)". I suspect that others will have their own licences. Creative Commons is NOT public domain. I can imagine the authors frustration that having recommended phylopic I'm now criticising the way they've used it, and I realise that this may come across as pedantry (it is). Sadly Creative Commons licensing is a bit of a mess for this very reason.

But I suggest that the authors look at each image's conditions of use, and if it requires attribution then the name of the person who created the image is added to Acknowledgements. If this seems excessive (it would be tedious, but the illustrators have asked to be appropriately credited), then at least remove the "public domain" statement as that is false.

Reviewer 4 ·

Basic reporting

Clear and understandable language, easy to read, and well-structured.

Some minor comments on detail:

• 116/117 refers to "raw data downloaded from GBIF", while line 104 mentioned downloading GBIF-interpreted data. Assuming that the paragraph still refers to the same download of interpreted data, suggest rephrasing 116 to "(...) first filtered the data as downloaded from GBIF ("raw", hereafter) (...)", to avoid misunderstandings around "raw" (unprocessed, original) data that are also available from GBIF.
• 242/243: the same link is duplicated. Should the second one point at a different URL?
• 266: "(...) that between one to half of the occurrence records (...)": for clarification: is it between one record and half of all records, or is some other word missing?
• Fig.2: suggest to consider labeling map axes in degrees lat/long or decimal degrees
• Fig.2, title/legend: suggest to be more explicit in the legend to signify absolute numbers of occurrence records removed/flagged in filtering
• Fig.3, legend: see Fig 2, above
• Fig.4: suggest to consider adding the total number of species (n= ) for each of A-D, assuming that the total size of the group matters to number of species removed

Experimental design

An interesting design to compare the impacts of a uniform approach across diverse species groups, highlighting varying importance of different data fields for diverse taxonomic groups. Clear questions outlined at the beginning.

With the added rationale for not making use of the issue flags already available from GBIF data processing (110-114), the study goal now appears somewhat ambiguous between an evaluation of GBIF-mediated data in particular in their suitability for biodiversity analyses and required preparatory work (e.g. 266ff, title) on one hand, and the evaluation of the importance and impact of specific data filters to be applied to any kind of database used in such analyses, with GBIF unfiltered data just chosen as a case study (112-114), on the other. Could this be clarified?

Validity of the findings

The findings are plausible and well-documented. The study provides a welcome perspective on the caveats and value of using data aggregated from a multitude of contributing datasets in environmental research, testing against an established, expert-driven process.

Additional comments

Very minor comments (side notes, feel free to ignore):
• 233: the terms "extinct" and "fossil" are not necessarily always coupled, since a species may be more recently extinct, with specimen (not fossil) vouchers
• 274: "to allow users to provide (...) feedback for particular records": technically, this is already possible. It is rather the visibility and rapid re-integration of feedback that is an issue.
• Tab.2, Biodiversity institutions: "or because they represent individuals from captivity or horticulture": that would be expected and by design. Following the Darwin Core standard, such records should be identifiable by a basis of record of "living"
• Tab 2, Individual count: "number of recorded individuals is 0 (record of absence)": some database management systems default to 0 for numerical values; other measures for status (present/absent) or abundance (% coverage) can help to disambiguate

---

## Round 0.3 · Minor Revisions

Thank you for the responses to the three referees. Regarding the last point by the third referee, I agree with them that Table 2 is a bit incorrect regarding individual count being zero. Surely if there is a record it must be at least one? What I understand the "individual count" field to means is if abundance was recorded which often is not the case. It is very common for database systems to place a zero in an empty numerical field. This can be checked by looking at other records for the same data set to see if this field always contains zero. It is probably exceptional that the present definition in Table 2 is true [i.e., "Records may be unsuitable if the number of recorded individuals is 0 (record of absence) or if the count is too high, as this is often related to records from barcoding or indicative of data entry problems."] Can you please check this again and consider rewording it because it may give users the idea that a failure to record abundance data means a record is questionable.

---

## Round 0.4 · accepted · Accept

Thank you for the good answer to my question. Thank you for choosing PeerJ.